# Metabolic Diversity of Xylariaceous Fungi Associated with Leaf Litter Decomposition

**DOI:** 10.3390/jof8070701

**Published:** 2022-07-01

**Authors:** Kohei Tabuchi, Dai Hirose, Motohiro Hasegawa, Takashi Osono

**Affiliations:** 1Graduate School of Science and Engineering, Doshisha University, Kyotanabe, Kyoto 610-0394, Japan; tabutabu0315@gmail.com; 2School of Pharmacy, Nihon University, Funabashi, Chiba 274-8555, Japan; hirose.dai@nihon-u.ac.jp; 3Faculty of Science and Engineering, Doshisha University, Kyotanabe, Kyoto 610-0394, Japan; mohasega@mail.doshisha.ac.jp

**Keywords:** biogeography, Biolog EcoPlate^TM^, functional diversity, physiological profiling, variation partitioning, Xylariaceae

## Abstract

Fungi in the family Xylariaceae are primary agents of leaf litter decomposition. However, the diversity of carbon source utilization by xylariaceous fungi and the relative effects on this from environmental and phylogenetic factors are largely unknown. This study assessed the metabolic diversity and redundancy of xylariaceous fungi, associated with leaf litter decomposition, by measuring their in vitro capacity to utilize multiple carbon sources. The work identified the relative influences of geographic and climatic sources, as well as the taxonomic and phylogenetic relatedness, of the fungi. Using Biolog EcoPlate^TM^, 43 isolates belonging to *Nemania*, *Xylaria*, *Nodulisporium*, *Astrocystis*, and *Hypoxylon,* isolated from *Castanopsis sieboldii* leaf litter at eight sites in Japan, were found to have the capacity to utilize a variety of carbohydrates, amino acids/amines, carboxylic acids, and polymers. The genera of xylariaceous fungi and their origins significantly affected their metabolic diversity and utilization of carbon sources. Variation partitioning demonstrated that dissimilarities in carbon utilization among fungal isolates were mostly attributable to site differences, especially climatic factors: mean annual temperature and precipitation, and maximum snow depth. Moreover, xylariaceous isolates that originated from adjacent sites tended to have similar patterns of carbon source utilization, suggesting metabolic acclimation to local environmental conditions.

## 1. Introduction

Fungi are the primary agents in the decomposition of leaf litter in terrestrial ecosystems [1,2]. Fungal hyphae that grow apically have the capacity to penetrate dead leaf tissues and excrete a variety of extracellular enzymes that catalyze the degradation of organic chemical constituents in the litter [3,4]. A taxonomically diverse array of fungi contributes to decomposition, including species of Ascomycota, Basidiomycota, and Mucoromycotina. Each fungal species possesses a different suite of extracellular enzymes involved in polymer degradation and has different capacities to metabolize carbon compounds [5]. Consequently, evaluating not only the richness and composition of fungal species but also their metabolic diversity and redundancy in terms of their utilization of carbon sources, such as carbohydrates, carboxylic acids, polymers, amino acids, and amines, is crucial for understanding the functional role of fungi in decomposition processes.

Fungi in the family Xylariaceae (Xylariales, Sordariomycetes, Ascomycota) are major components of the mycobiota in terrestrial ecosystems [6,7] and serve as decomposers [8,9,10,11], endophytes [12,13,14], pathogens [15,16], and symbionts with termite nests [17,18]. Their importance in leaf litter decomposition lies in the ligninolytic and cellulolytic activity causing the soft rot (type II) of plant cell walls [19,20]. Accordingly, studies on the metabolic activity of xylariaceous fungi have addressed the production of ligninolytic peroxidases, cellulases, and secondary metabolites [21,22,23,24]. Nevertheless, little is known about the diversity and redundancy of carbon source utilization by xylariaceous fungi and the relative contributions of environmental and phylogenetic factors. To the knowledge of the authors, few studies have explored the environmental and evolutionary signals of carbon source utilization using cultivable isolates of fungi and other prokaryotic microbes [25]. The hypothesis tested in the present study was that key ecological and phylogenetic features of fungi (i.e., patterns of geographic distribution and evolutionary history) can predict their metabolic diversity.

The aims of the present study were to (i) elucidate the metabolic diversity and redundancy of xylariaceous fungi, associated with leaf litter decomposition, by measuring in vitro activity drawing on multiple carbon sources and (ii) to disentangle the relative effect of climatic conditions and geographic locations of the sites from which the fungi originated, as well as their taxonomic and phylogenetic relatedness. We isolated xylariaceous fungi that took part in the decomposition of the leaf litter of a single host tree, *Castanopsis sieboldii* (Makino) Hatusima ex Yamazaki et Masiba, collected at eight geographically distinct study sites with different climatic conditions in temperate regions of Japan. We chose *C. sieboldii* as the focal host species because this tree is widely distributed across several climatic regions in the Japanese archipelago [26], and xylariaceous fungi frequently occur in the leaf litter [27]. Fungal isolates were used for molecular phylogenetic analyses in annotating their taxonomic and phylogenetic positions and were tested for their physiological capacity to utilize the 31 carbon compounds assayed by Biolog EcoPlates^TM^. By applying nonmetric multidimensional scaling (NMDS) and variation partitioning, we visualized the dissimilarities in the physiological profiles of xylariaceous fungi and identified the relative contributions to these of climatic, geographic, and taxonomic/phylogenetic factors.

## 2. Materials and Methods

### 2.1. Study Site and Fungal Isolation

Samples were collected at eight study sites in temperate regions of Japan (30.257 to 35.176° N, 130.585 to 140.120° E, 94 to 347 m a.s.l.) (Table 1, Figure 1). The study sites were located in mature, evergreen, broad-leaved forests dominated by *Castanopsis sieboldii*. Fieldwork, specifically leaf litter collection, was undertaken once at each site from 2009 to 2011. Collection was conducted in the summer (from July to September) when the occurrence of bleached portion was most evident on leaf litter that had fallen in the previous spring, indicative of the active lignin decomposition by fungi [5]. The leaf-sampling procedures were previously detailed [27]. At each site, we randomly set 10 quadrats (15 × 15 cm) at least 3 m from each other. A total of 20 leaves of *C. sieboldii* with evident bleaching, of which more than half of the original leaf area remained, were collected from the surface of the forest floor within the quadrats, with 2 leaves per quadrat. These leaves were placed in paper bags and preserved at 4 °C for no longer than 3 days before the isolation of fungi. Fungi were isolated from leaf disks excised from the bleached portions of fallen leaves, using the surface disinfection method previously described [28]. Putative xylariaceous fungi that produced conidia and conidiophores of anamorphic Xylariaceae on the plates, such as *Xylocoremium*, *Geniculosporium*, and *Nodulisporium*, and/or dark pseudosclerotinial plates in submerged hyphae, were subcultured to establish pure cultures and used in the molecular analysis described in the following section.

### 2.2. Molecular Analysis

The rDNA internal transcribed spacer (ITS) region of fungal isolates was analyzed using primers ITS1f/LR3, according to the methods previously described [27]. Using BLAST+ [29], the determined ITS sequences were compared with the rDNA sequences available in the GenBank database and taxonomically assigned using the query-centric auto-k-nearest-neighbor (QCauto) method [30] as implemented in Claident (v. 0.2.2019.05.10), a phylogeny-free method (Appendix A). A benchmark analysis showed that the combination of the QCauto and life cycle assessment algorithms was less susceptible to erroneous database sequences and returns the most accurate taxonomic identification results among the existing methods of automated DNA metabarcoding [30]. Data from 43 fungal isolates belonging to Xylariaceae were analyzed in this study, and their sequences were deposited in the DNA Data Bank of Japan (DDBJ) (LC505086-LC505334; Appendix A).

### 2.3. Phylogenetic Analysis

To estimate the effects of the phylogenetic relations of the fungal isolates on their carbon utilizations, we constricted the phylogenetic tree and then calculated Moran’s eigenvector maps (MEMs) to represent phylogenetic distance vectors (denoted here as pMEMs) according to the method previously described [31]. First, in the ITS region of all isolates, a multiple alignment was generated using MAFFT software [32] on the ITS sequences. A maximum-likelihood phylogenetic tree was estimated using RAxML-NG v. 1.0.0 [33] under a GTRGAMMA (GTR + G + I) model. Maximum-likelihood bootstrap percentages and a tree were obtained after 500 bootstrap replicates, followed by a search for the most likely tree. The pairwise phylogenetic distances of each isolate, calculated during the construction of the phylogenetic tree, were then used to calculate the pMEM vectors with the vegan and adespatial packages of R (v. 4.0.3). The pMEM vectors are orthogonal; they represent the amount of phylogenetic variation proportional to their eigenvalue and are listed in order of decreasing explained variance [31]. The nine pMEM vectors (denoted as pMEM1 to pMEM9; Appendix A) with positive values of Moran’s I were used in the subsequent statistical analysis. A phylogenetic tree was generated to visualize the variability of base sequences of rDNA ITS among the fungal isolates and to generate pMEM vectors. It is not comprehensive nor meant to more broadly illustrate phylogenetic relationships.

### 2.4. Carbon Source Utilization

The metabolic potential of 43 xylariaceous isolates was evaluated using Biolog EcoPlate^TM^ with 31 different carbon sources (Biolog Inc., Hayward, CA, USA). Fungal isolates were incubated in 50 mL of 1% (*w*/*v*) malt extract solution and incubated in the dark for 3 weeks at 20 °C. Mycelia in the solution were homogenized in a blender at 10,000 r.p.m. for 3 min. A suspension was made by adding 5 mL of the homogenate into 45 mL of sterilized, distilled water, and a series of dilutions was prepared. After that, 100 μL of a 10^2^ dilution was aseptically transferred to each well (31 carbon sources + 1 negative control) of an EcoPlate^TM^. Three replicated wells in a single plate were used for each fungal isolate. The plates were incubated in the dark at 20 °C for 14 days, and the resultant color development was measured at 595 nm using a plate reader (Infinite F50R; Tecan Co., Kanagawa, Japan).

The color development in the negative control well was subtracted from each of the 31 carbon source wells to obtain the optical density (OD) for each of the carbon sources. OD values less than 0.0001, the detection limit of the plate reader, were regarded as a negative substrate utilization result. The activity of fungal isolates to utilize carbon sources was quantitatively summarized using the average well color development (AWCD): AWCD = Σ OD*i*/31, where OD*i* is the OD in the well of *i*th substrate. Simpson’s diversity index (D) and equitability (E) were calculated using D = 1/Σ P*i*^2^ and E = D/S, where P*i* is a proportion of the OD*i* over the sum of the ODs of all 31 substrates in a plate, and S is the number of wells with a positive substrate utilization (i.e., the richness of substrate utilization). The 31 substrates were classified into four chemical groups as previously described [34]: carbohydrates (ten substrates), amino acids/amines (eight substrates), carboxylic acids (nine substrates), and polymers (four substrates) (Appendix A).

### 2.5. Statistical Analyses

We prepared a datasheet of Biolog profiles recording the ODs of the 31 substrates for all fungal isolates. Data of mean air temperature (MAT; °C), mean annual precipitation (MAP; mm), and maximum snow depth (MSD; cm) were obtained from the nearest station to each study site from the Automatic Metrological Data Acquisition System (Japan Meteorological Agency).

We used generalized linear models (GLMs) to examine the effects of genus, site, and the interaction of genus × site on the four Biolog indices (AWCD, the richness of substrate utilization, Simpson’s D, and equitability) and on the mean ODs of four chemical groups (carbohydrates, amino acids/amines, carboxylic acids, and polymers). The error structure of the GLM was Gaussian. The glm function in the R (v. 3.6.2) software was used to perform the analysis.

We used NMDS with Euclidean distance metric to analyze the variation among Biolog profiles. The NMDS analysis was conducted using the metaNDS function and the default settings of the vegan package of R software [35]. The “envfit” command of the vegan package applied permutation tests (999 permutations) to examine the correlation of NMDS structure with genus and site, and with the geographical and climatic attributes of the sites.

We used variation partitioning based on redundancy analysis (RDA, “rda” and “varpart” commands in the vegan package) to quantify the contributions of site, climate, spatial variables, and taxonomy + phylogeny to the Biolog profiles of the xylariaceous fungal isolates. The relative weight of each fraction (pure, shared, and unexplained) was estimated following the methodology previously described [36]. For the RDA, the OD data for each isolate and for each substrate were converted into a Euclidean distance matrix and used to create four models, based on site, climate, spatial factors, and taxonomy + phylogeny. Details of the variation partitioning method were previously described [37]. The full set of climatic, spatial, and taxonomy + phylogeny models included as factors: MAT + MAP + MSD; three spatial MEMs (denoted here as sMEMs); and taxonomy (i.e., genus) plus nine pMEM vectors. A spatial model, sMEM, was calculated in the same way as pMEM, described above, but based on the coordinates of each study site. For each of these four models, we then conducted forward selection (999 permutations with an alpha criterion of 0.05) to select variables that significantly influenced dissimilarities in carbon source utilization [38]. Based on these four models, we performed variation partitioning by calculating adjusted the R^2^ values for each fraction [36].

## 3. Results

### 3.1. Taxonomic Assignment

The 43 xylariaceous isolates were assigned to five genera (Table 2). *Nemania* S.F. Gray was the most frequent genus with 26 isolates, followed by *Xylaria* Hill ex Schrank (10 isolates), *Nodulisporium* Preuss (5 isolates), and *Astrocystis* Berk. & Broome and *Hypoxylon* Bull. (1 isolate each) (Table 2). *Nemania* isolates were variable in their base sequences of rDNA ITS (Figure 2) but were all assigned to *N. diffusa* (Sowerby) Gray (Appendix A). Seven of the *Xylaria* isolates were not assigned to species but the other three were identified as *X. cubensis* (Mont.) Fr. (Appendix A). Isolates of *Nodulisporium* and *Astrocystis* were not assigned to species, whereas the *Hypoxylon* isolate was assigned to *H. vinosopulvinatum* Y.-M. Ju, J. D. Rogers & Hsieh. (Appendix A). Between 7 and 14 isolates were obtained from sites Az, Ko, Id, and Ch, whereas 1 isolate was obtained from each of the sites Ya, My, Tu, and Ot (Table 2).

### 3.2. Carbon Source Utilization Patterns

The AWCD of the 43 xylariaceous isolates ranged from 0.003 to 0.425, and the mean AWCD values were significantly different among the genera and sites (Figure 3a; Table 3). The richness of substrate utilization ranged from 11 to 31 and was significantly different between sites, with the interaction of genus × site being significant (Figure 3b; Table 3). The Simpson’s D ranged from 1.5 to 18.8 and was significantly different among both genera and sites (Figure 3c; Table 3). The equitability ranged from 0.05 to 0.93 and was also significantly different among the genera and sites (Figure 3d; Table 3). In general, the diversity of carbon source utilization was greater in *Nodulisporium* than in the other genera (Figure 3). For all four Biolog indices, the effect of site was generally greater than that of genus (Table 3).

All 43 xylariaceous isolates were positive in the utilization of Tween 40 (polymer) and D-xylose (carbohydrate), whereas 22 isolates in the utilization of phenylethyl-amine (amine), 25 isolates each in D-galactonic acid γ-lactone (carbohydrate) and 2-hydroxy benzoic acid were positive (carboxylic acid) (Appendix A). The mean number of positive isolates was greater in polymers (40.3 isolates) than in carbohydrates (32.3 isolates), carboxylic acids (32.2 isolates), and amino acids/amines (31.3 isolates) (Appendix A).

Among the 43 xylariaceous isolates, the mean ODs of carbohydrates ranged from 0.003 to 0.530 (Figure 4a), amino acids/amines from 0.000 to 0.379 (Figure 4b), and carboxylic acids from 0.000 to 0.211 (Figure 4c). For each of these, the mean ODs were significantly different among both genera and sites. The mean ODs of polymers ranged from 0.001 to 0.829 and were significantly different between sites but not between genera (Figure 4d). In general, the ODs of carbohydrates, amino acids/amines, and carboxylic acids were greater in *Nodulisporium* than in the other genera (Figure 4). Again, the effect of site was generally greater than that of the genera for all four chemical groups (Table 3).

### 3.3. Partitioning the Dissimilarity of Carbon Source Utilization

The NMDS ordination revealed separation of Biolog profiles among both genera and sites (Figure 5). The ordination was significantly affected by site (“envfit” function; R^2^ = 0.651, *p* < 0.001), but was not significantly affected by genus (R^2^ = 0.160, *p* = 0.186). The ordination was significantly related to site longitude (“envfit” function; R^2^ = 0.303, *p* < 0.001), elevation (R^2^ = 0.188, *p* = 0.009), and MSD (R^2^ = 0.181, *p* = 0.012).

In the variation partitioning (Figure 6), MAT, MAP, and MSD were selected as climatic factors, and two sMEM vectors (vectors 3 and 1) were selected as spatial factors. In the taxonomy + phylogeny model, one pMEM vector (vector 1) was selected, whereas taxonomic factors (fungal genus) were not selected. Consequently, this can be referred to simply as a phylogeny model. The percentages explained by the site, climate, spatial, and phylogeny fractions were 39%, 31%, 21%, and 9%, respectively. The pure fractions of these four models were all 0%. Among the shared fractions, the greatest proportion of variation explained by site difference was attributed to the climate of the sites, secondly to the geographic distance between the sites, and finally to the phylogeny of the isolates. In total, 39% of the variation in Biolog profiles was explained, and the remaining 61% was unexplained.

## 4. Discussion

The above results reveal the diversity of metabolic activity among representative genera of Xylariaceae in utilizing carbon sources in leaf litter decomposition [39]. Our tests using the Biolog EcoPlate^TM^ revealed the diversity of substrate utilization by xylariaceous fungi and the redundancy arising from their capacity to utilize a suite of substrates, including carbohydrates, amino acids/amines, carboxylic acids, and polymers (Appendix A). The xylariaceous isolates generally exhibited a rapid and extensive utilization of polymers (including Tween 40, Tween 80, α-cyclodextrin, and glycogen) and carbohydrates (such as *N*-acetyl-D-glucosamine, D-mannitol, and D-cellobiose) (Figure 4). This suggests that these fungi can produce extracellular enzymes that break down these compounds, in addition to structural polymers in plant cell walls, such as cellulose and lignin [23,40]. Similarly, carboxylic acids and amino acids/amines may be used in the decomposing activity and secondary metabolism of xylariaceous fungi [24,41].

Both the genera of xylariaceous fungi and their geographic locations significantly affected their metabolic diversity and profiles of utilization of carbon sources (Table 3, Figure 5). The genus *Nodulisporium* included some isolates that have higher AWCD, Simpson’s D, and equitability of carbon source utilization than other genera, resulting in significantly greater mean values of these indices for *Nodulisporium* than for other genera (Figure 3). Interestingly, this contrasts to a prior study, under pure culture conditions, which found the capacity for the decomposition of leaf litter to be generally lower in fungal isolates of *Nodulisporium* than in those of *Xylaria* and *Nemania* [5]. However, the present study included only five isolates of *Nodulisporium*, and these were phylogenetically close to each other (Figure 2), making it difficult to validate the differences in carbon source utilization between genera. Similarly, caution is appropriate when interpreting the effect of the site of origin of isolates on carbon source utilization. Unfortunately, the number of isolates in genera and sites were not fully orthogonal because of inevitable bias in the sampling of fungal isolates. Therefore, the significant interaction of genus × site in explaining the richness of substrate utilization (Table 3) is largely attributable to the higher values for *Nemania* isolates at a particular site (i.e., site Az) compared with other sites (e.g., sites Id and Ch) (Appendix A).

To gain further insights into the relative importance of factors affecting carbon source utilization by xylariaceous fungi, we performed variation partitioning and demonstrated that dissimilarity in carbon source utilization was mostly attributable to the site differences and that much of this derived from climatic factors, such as MAT, MAP, and MSD (Figure 6). This result suggests a physiological adaptation, in terms of carbon source utilization, of xylariaceous fungi to local climates. Previous studies reported that metabolic activity and diversity examined with Biolog EcoPlate^TM^ could vary with temperature and moisture [42,43]. Snow cover has also been shown to affect the activity of fungi in leaf litter decomposition [44]. In addition, geographic distances between sites (i.e., spatial factors or sMEM) contributed to site differences, implying a spatial structure in carbon source utilization by xylariaceous fungi [45]. In other words, xylariaceous isolates originating from closer sites tended to exhibit more similar patterns of carbon source utilization. This was exemplified, as noted above, by the disparity between *Nemania* isolates at the Az site in the Shikoku district and those at the Id and Ch sites in the Kanto district. This localization of fungal carbon source utilization may be a consequence of the presence of locally abundant fungal species affected by such non-niche processes as priority effect and dispersal limitation [27,46,47]. Finally, phylogenetic distances between the isolates (pMEM) also contributed to site differences in the Biolog profiles, indicating that phylogenetically close isolates can share similar patterns of carbon source utilization. The lower explanatory power of the phylogeny model indicated that climate and site location have the greater influence on carbon source utilization by xylariaceous fungi, and this suggests metabolic acclimation to local environmental conditions.

In the variation partitioning, approximately 61% of the variation remained unexplained (Figure 6). There may be two reasons for this unexplained variation. Firstly, the number of sites and fungal isolates may not have been large enough to detect additional potential variations in carbon source utilization by xylariaceous fungi. Secondly, unmeasured environmental and/or genomic factors, such as the physiological and chemical properties of the leaf litter, the copy number of genes coding for enzymes, and biotic interactions with other fungi and/or soil organisms, are all plausible influences on variation [48,49,50]. Nonetheless, we believe that the present study is the first to use the Biolog EcoPlate^TM^ to elucidate the metabolic diversity of multiple fungal isolates, of a particular taxonomic group sourced from different sites and to explore the effects of environmental, spatial, and phylogenetic factors. Further studies are needed to test whether the results of the present study are also applicable to other taxonomic groups of fungi, to fungi on leaf litter of different tree species, and to different geographic and climatic regions.

## Figures and Tables

**Figure 1 jof-08-00701-f001:**
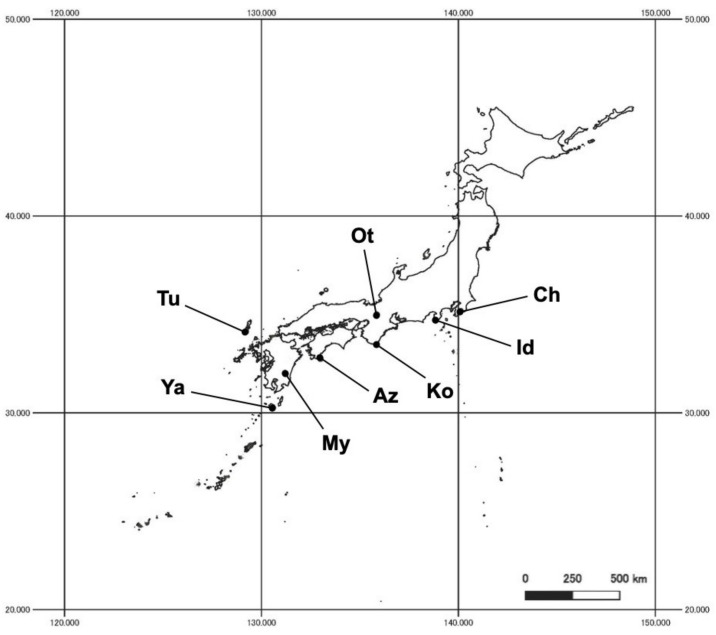
Locations of eight study sites in Japan. Site codes are as in Table 1.

**Figure 2 jof-08-00701-f002:**
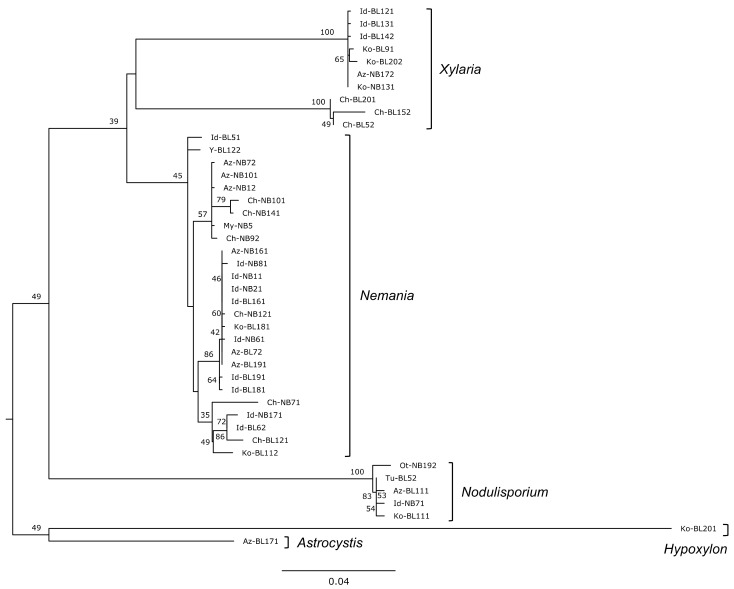
Maximum-likelihood tree based on the rDNA ITS sequences of 43 xylariaceous isolates used in the present study. Names of identified xylariaceous genera are depicted in brackets. Bootstrap values above 30% are given adjacent to the corresponding node.

**Figure 3 jof-08-00701-f003:**
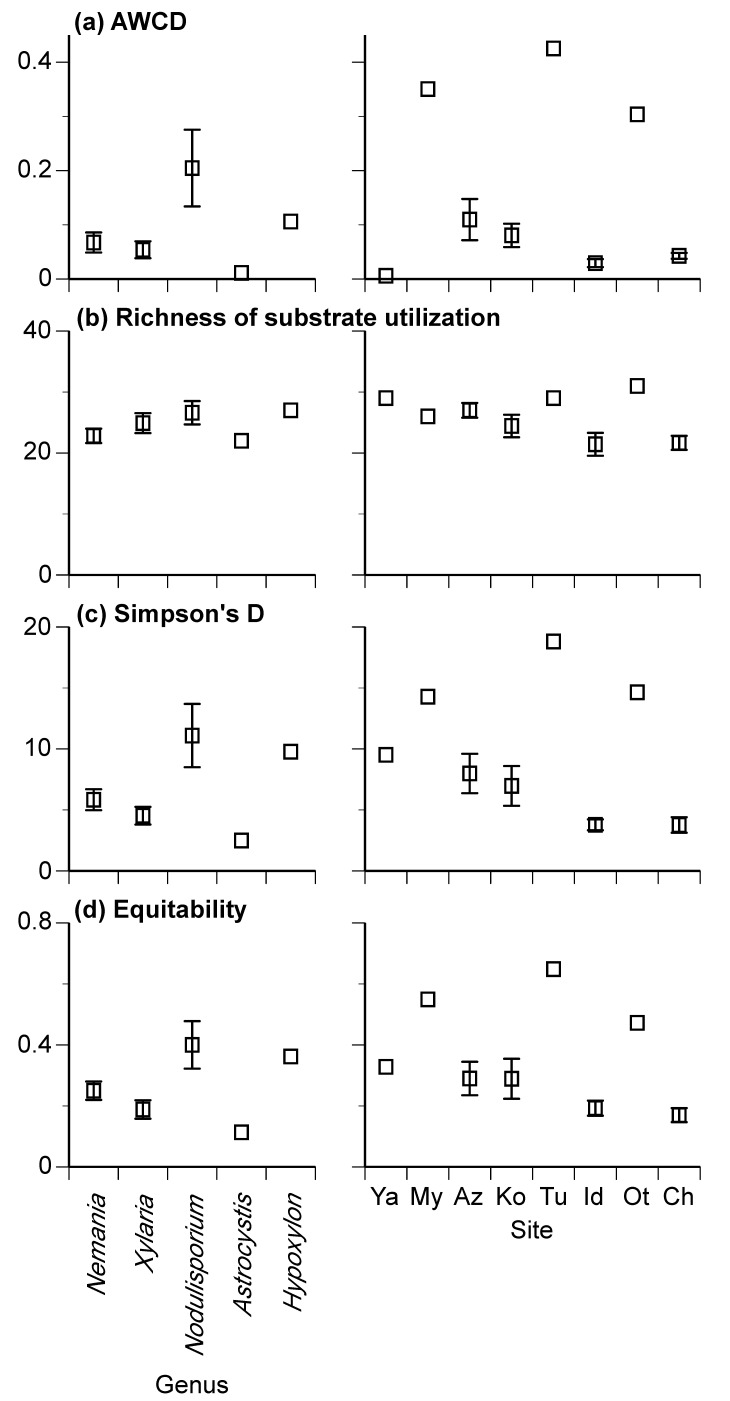
Average well color development (AWCD) (**a**), the richness of substrate utilization (**b**), Simpson’s D (**c**), and equitability (**d**) for 43 xylariaceous isolates. Mean values are indicated for genera and site. Bars indicate standard errors. Site codes are shown in Table 1.

**Figure 4 jof-08-00701-f004:**
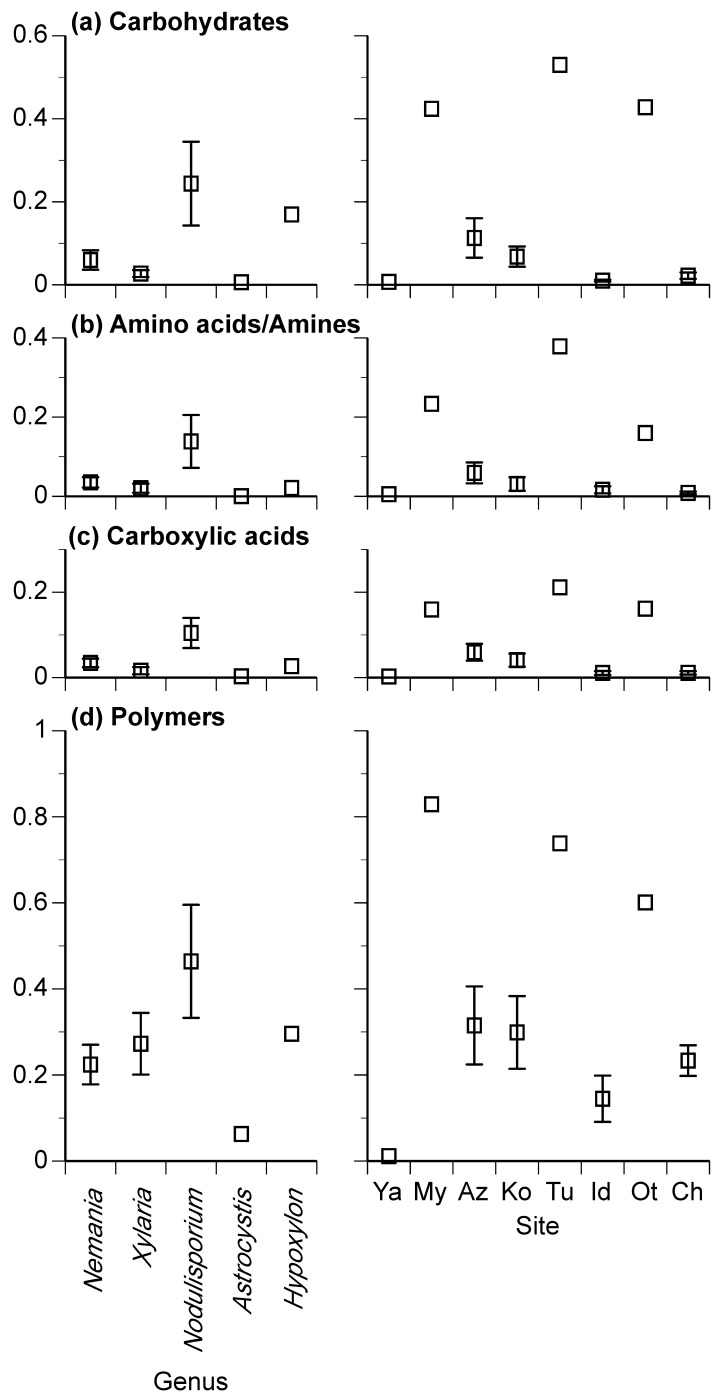
Mean optical density of carbohydrates (**a**), amino acids/amines (**b**), carboxylic acids (**c**), and polymers (**d**) for 43 xylariaceous isolates. Mean values are indicated for genera and site. Bars indicate standard errors. Site codes are shown in Table 1.

**Figure 5 jof-08-00701-f005:**
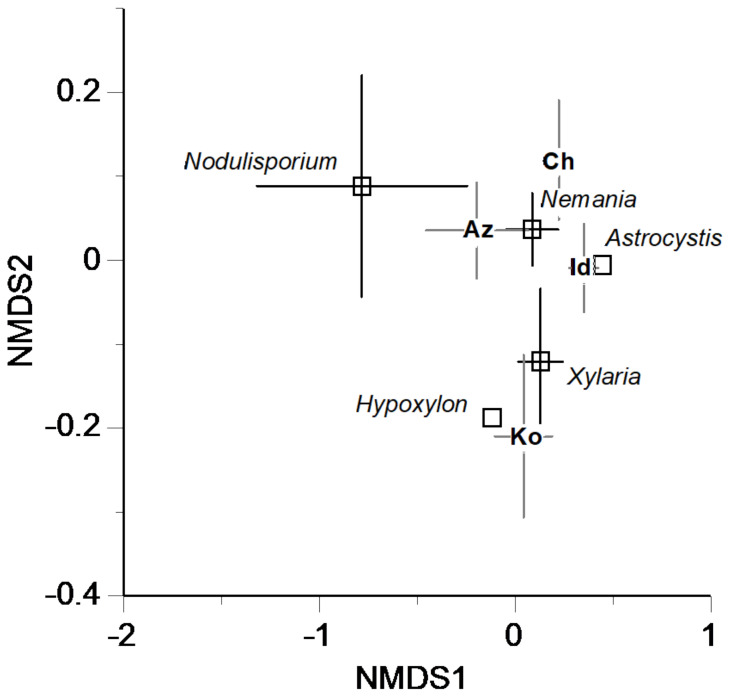
Dissimilarity among the Biolog profiles for 43 xylariaceous isolates as revealed by NMDS ordination using Euclidean distance (stress value = 0.066). Mean values are indicated for genera and site. Bars indicate standard errors. NMDS scores of four sites [Ya (0.49, 0.02), My (−2.05, −0.43), Tu (−2.34, 0.02), Ot (−1.57, 0.60)] from which only one isolate was obtained are not shown. Site codes are shown in Table 1.

**Figure 6 jof-08-00701-f006:**
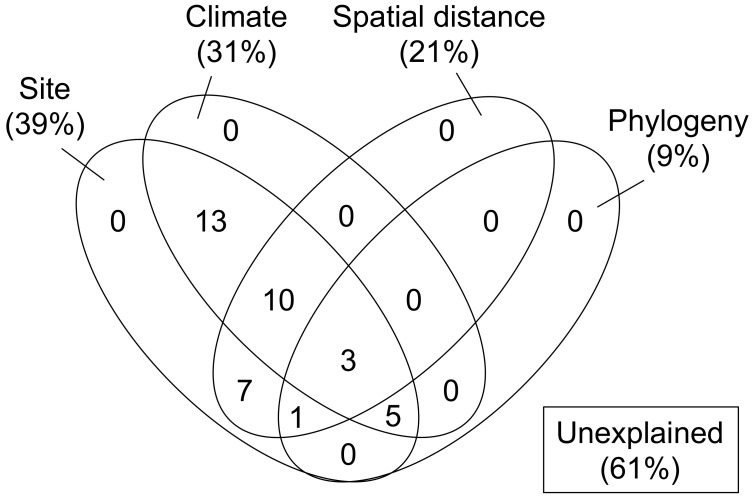
Benn diagram showing the pure and shared effects of site, climate, space, and phylogeny on the Biolog profiles for 43 xylariaceous isolates as derived from variation-partitioning analysis. Numbers indicate the proportions of explained variation.

**Table 1 jof-08-00701-t001:** Location of study sites, climates, and sampling of 43 xylariaceous isolates. The eight sites are ordered according to their latitudes. Lat, latitude; Long, longitude; Elev, elevation; MAT, mean annual temperature; MAP, mean annual precipitation; MSD, mean snow depth.

Site Code	Site Name	Lat (°N)	Long (°E)	Elev (m)	MAT (°C)	MAP (mm)	MSD (cm)	Sampling Month and Year
Ya	Yakushima Is., Kagoshima	30.257	130.585	195	19.1	3589.8	0	Aug 09
My	Aya, Miyazaki	32.027	131.194	241	15.4	2838.8	0	Sept 09
Az	Cape Ashizuri, Kochi	32.740	132.999	271	16.9	2458.8	0	Jul 10
Ko	Kii-Oshima, Kyoto Univ., Wakayama	33.466	135.833	94	17.0	2547.1	0	Aug 11
Tu	Mt. Taira, Tsushima Is., Nagasaki	34.142	129.218	347	14.0	2122.7	0	Sept 09
Id	Jugei, Univ. Tokyo, Izu Pen., Shizuoka	34.691	138.839	135	14.9	2042.9	2	Aug 11
Ot	Mt. Nagara, Otsu, Shiga	35.004	135.857	130	14.3	1496.2	0	Jul 10
Ch	Kiyosumi, Univ. Tokyo, Chiba	35.176	140.120	195	13.5	2075.7	5	Aug 11

**Table 2 jof-08-00701-t002:** Genera and number of xylariaceous isolates obtained from eight study sites. Site codes are as in Table 1.

Genus	Ya	My	Az	Ko	Tu	Id	Ot	Ch	Total
*Nemania*	1	1	6	2	0	10	0	6	26
*Xylaria*	0	0	1	3	0	3	0	3	10
*Nodulisporium*	0	0	1	1	1	1	1	0	5
*Astrocystis*	0	0	1	0	0	0	0	0	1
*Hypoxylon*	0	0	0	1	0	0	0	0	1
Total	1	1	9	7	1	14	1	9	43

**Table 3 jof-08-00701-t003:** Summary of generalized linear model for average well color development (AWCD), the richness of substrate utilization, Simpson’s D, equitability, and mean optical density of carbohydrates, amino acids/amines, carboxylic acids, and polymers, using genus, site, and their interaction as predictor variables. *** *p* < 0.001, ** *p* < 0.01, * *p* < 0.05, ns not significant.

Responsible Variable	Genus			Site			Genus × Site		
	Deviance	*p*		Deviance	*p*		Deviance	*p*	
**Indices**									
AWCD	0.09	0.000	***	0.22	0.000	***	0.01	0.855	ns
Richness of substrates utilization	90.4	0.372	ns	333.2	0.028	*	335.6	0.007	**
Simpson’s D	177.7	0.001	***	347.2	0.000	***	74.2	0.155	ns
Equitability	0.18	0.021	*	0.31	0.006	**	0.08	0.420	ns
**Chemical groups**									
Carbohydrates	0.19	0.000	***	0.40	0.000	***	0.01	0.764	ns
Amino acids/Amines	0.05	0.000	***	0.14	0.000	***	0.02	0.052	ns
Carboxylic acids	0.029	0.000	***	0.059	0.000	***	0.002	0.904	ns
Polymers	0.28	0.181	ns	0.79	0.015	*	0.22	0.436	ns

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
