# Peer review of "Metabolic Diversity of Xylariaceous Fungi Associated with Leaf Litter Decomposition"

_jof, 2022, doi:10.3390/jof8070701_

Round 1

Reviewer 1 Report

Minor revisions and comments are highlited in PDF manuscript;

Major comment: with the analysis of the ITS region, many isolates were not identified at the species level. Since each strain has differences in the expression of enzyme-coding genes and this contributes to the differences in carbon utilization, I suggest using more specific primers such as beta tubulin and calmodulin to identify isolates at the species level to detect differences in carbon utilization within each genus. This could also help clarify why 61% of variation remained unexplained.

Author Response

Answers to the comments by Reviewer 1

Minor revisions and comments are highlited in PDF manuscript;

Reply:

Thank you for helpful comments to the manuscript. Your revisions are incorporated in the revised manuscript. The names of fungal genera in Figures 3 and 4 were awful in PDF file, but I note that these are correctly displayed in Word file; the defeat is probably due to the file conversion on the website.

Major comment: with the analysis of the ITS region, many isolates were not identified at the species level. Since each strain has differences in the expression of enzyme-coding genes and this contributes to the differences in carbon utilization, I suggest using more specific primers such as beta tubulin and calmodulin to identify isolates at the species level to detect differences in carbon utilization within each genus. This could also help clarify why 61% of variation remained unexplained.

Reply:

We agree with your comment that using more specific primers like beta tubulin and calmodulin in addition to the ITS region would provide higher resolution of species identification of fungal isolates. In the present study, we utilize the ITS region firstly because this is the common barcoding region, leading to the successful species identification of a reasonable number of isolates (30 isolates). More importantly, the phylogenetic vectors derived from the variation in the ITS region were significantly related to the variation in enzyme activity (Figure 6). We thus believe that the ITS region is an appropriate genetic marker for the purpose of the present study.

Takashi Osono

Reviewer 2 Report

The manuscript submitted for review is written correctly and clearly. The title is short and pithy. The abstract is correctly spelt. Meaningful keywords. They will make it possible to find publications in the future. They don't repeat themselves with the title. The introduction is concise and general. The problem of decomposition of leaves and the participation of fungi in this process has not been fully presented. The introduction is dominated by hypotheses and the research goal, not an introduction to the topic. The samples taken for analysis were leaves from deciduous forests. What kinds of trees? Did the analyzed leaves always belong to C. sieboldii? By default, I will answer this question. I propose to start the description of the research material with this information. Why were the leaves taken from July to September? Were the harvested leaves freshly fallen? There is no information in the methodological description. Were outgroups used in the phylogenetic analysis? Results and discussion wrote correctly. However, reading the manuscript (especially the introduction and discussion) gives the impression that the author is not entirely familiar with the topic under study.

Author Response

Answers to the comments by Reviewer 2

The manuscript submitted for review is written correctly and clearly. The title is short and pithy. The abstract is correctly spelt. Meaningful keywords. They will make it possible to find publications in the future. They don't repeat themselves with the title. The introduction is concise and general. The problem of decomposition of leaves and the participation of fungi in this process has not been fully presented. The introduction is dominated by hypotheses and the research goal, not an introduction to the topic.

Reply:

Thank you for critical comments to the manuscript and positive evaluation of Introduction section. The background information on the roles of fungi in decomposition processes of leaf litter is described in the first paragraph of Introduction, and in the second one what is not clarified yet in previous works is specified.

The samples taken for analysis were leaves from deciduous forests. What kinds of trees? Did the analyzed leaves always belong to C. sieboldii? By default, I will answer this question. I propose to start the description of the research material with this information. Why were the leaves taken from July to September? Were the harvested leaves freshly fallen? There is no information in the methodological description. Were outgroups used in the phylogenetic analysis?

Reply:

Leaf samples were collected in evergreen forests, not in deciduous forests, and all leaves collected are those of one tree species: Castanopsis sieboldii, as are described in the first paragraph of Materials and Methods. We expand the description of research material in that paragraph of the revised manuscript. Collection was conducted from July to September when the occurrence of bleached portion was most evident on leaf litter fallen in the previous spring, indicative of the active lignin de-composition by fungi. This is stated in the revised manuscript. We used Hypoxylon and Astrocystis as outgroups for constructing the phylogenetic tree.

Results and discussion wrote correctly. However, reading the manuscript (especially the introduction and discussion) gives the impression that the author is not entirely familiar with the topic under study.

Reply:

Thank you for positive evaluation of Introduction and Discussion sections. We have been studying the ecology of fungi associated with leaf litter decomposition for 25 years, but yes, we might still be unfamiliar with this topic.

Takashi Osono

Reviewer 3 Report

This paper entitled"Metabolic Diversity of Xylariaceous Fungi Associated with Leaf Litter Decomposition" is appeared to be a nice piece of work and will provide more information and reference for future study. However, I think this need reject in the present form. The main problem are as follows:

1. The phylogenetic analysis of 43 strains was so cruded, which could affect subsequent analysis results. According to the phylogenetic tree, we can't get the results described by the author, and this will have a significant impact on the results of this paper.

2. The generic names and species names appear for the first time in the article, it is necessary to add a designated person.

3. line 111, whether 500 boostrap replicates is  it appropriate

4. Figure 3 and 4 are unfriendly to readers, because of the unclear drawing notes

Author Response

Answers to the comments by Reviewer 3

This paper entitled"Metabolic Diversity of Xylariaceous Fungi Associated with Leaf Litter Decomposition" is appeared to be a nice piece of work and will provide more information and reference for future study. However, I think this need reject in the present form. The main problem are as follows:

  1. The phylogenetic analysis of 43 strains was so cruded, which could affect subsequent analysis results. According to the phylogenetic tree, we can't get the results described by the author, and this will have a significant impact on the results of this paper.

Reply:

Thank you for useful comments to the manuscript. We used the variation of ITS base sequences to calculate phylogenetic vectors to represent phylogenetic distances among the 43 isolates, because the ITS regions contain reasonable variations for the purpose of the present study. Constructing the phylogenetic tree was mandatory for calculating phylogenetic vectors. We noted in Lines 130-133 that the phylogenetic tree was generated to visualize the variability of base sequences of rDNA ITS among the fungal isolates and to generate phylogenetic vectors and that the tree is not comprehensive nor meant to illustrate phylogenetic relationships more broadly. Therefore, we believe that the phylogenetic tree is crucial for the results we described in this paper.

  1. The generic names and species names appear for the first time in the article, it is necessary to add a designated person.

Reply:

Author names are added to the names of genus and species when appeared first time in the manuscript.

  1. line 111, whether 500 boostrap replicates is it appropriate

Reply:

We believe this is appropriate as we confirmed the stable likelihood and the robust tree.

  1. Figure 3 and 4 are unfriendly to readers, because of the unclear drawing notes

Reply:

The names of fungal genera in Figures 3 and 4 were awful in PDF file, but I note that these are correctly displayed in Word file; the defeat is probably due to the file conversion on the website.

Takashi Osono

Round 2

Reviewer 1 Report

Thank you for clearing up my doubts. Just pay attention to figures 3 and 4 which continue to be formatted incorrectly. Once these have been corrected, I believe that the work can be considered for publication.

Reviewer 2 Report

The manuscript submitted for re-review has been corrected in line with the comments. I only have two minor (editor) comments regarding figures 3 and 4. I have marked it in pdf. After correcting, I believe that the manuscript can be accepted for publication.

Reviewer 3 Report

The quality of the revised paper was limited improvement. I would like to suggest the Editor to reject in the present form. The main problems were as follows: 

1. As the basis for studying the metabolic diversity of this kind of fungi, species identification is particularly important. Unfortunately, the work in this article is obviously not enough.

2. The figure 3 and 4 were not revised.